# Formation of β-Strand Oligomers of Antimicrobial Peptide Magainin 2 Contributes to Disruption of Phospholipid Membrane

**DOI:** 10.3390/membranes12020131

**Published:** 2022-01-21

**Authors:** Munehiro Kumashiro, Ryoga Tsuji, Shoma Suenaga, Koichi Matsuo

**Affiliations:** 1Department of Physical Science, Graduate School of Science, Hiroshima University, 1-3-1 Kagamiyama, Higashi-Hiroshima, Hiroshima 739-8526, Japan; kumashiro@hiroshima-u.ac.jp (M.K.); barian1212@gmail.com (S.S.); 2Physics Program, Graduate School of Advanced Science and Engineering, Hiroshima University, 1-4-1 Kagamiyama, Higashi-Hiroshima, Hiroshima 739-8527, Japan; m211517@hiroshima-u.ac.jp; 3Hiroshima Synchrotron Radiation Center, Hiroshima University, 2-313 Kagamiyama, Higashi-Hiroshima, Hiroshima 739-0046, Japan

**Keywords:** amyloid, antimicrobial peptide, β-strand oligomers, fluorescence anisotropy, linear dichroism, magainin 2, peptide-membrane interaction, synchrotron-radiation circular dichroism

## Abstract

The antimicrobial peptide magainin 2 (M2) interacts with and induces structural damage in bacterial cell membranes. Although extensive biophysical studies have revealed the interaction mechanism between M2 and membranes, the mechanism of membrane-mediated oligomerization of M2 is controversial. Here, we measured the synchrotron-radiation circular dichroism and linear dichroism (LD) spectra of M2 in dipalmitoyl-phosphatidylglycerol lipid membranes in lipid-to-peptide (L/P) molar ratios from 0–26 to characterize the conformation and orientation of M2 on the membrane. The results showed that M2 changed from random coil to α-helix structures via an intermediate state with increasing L/P ratio. Singular value decomposition analysis supported the presence of the intermediate state, and global fitting analysis revealed that M2 monomers with an α-helix structure assembled and transformed into M2 oligomers with a β-strand-rich structure in the intermediate state. In addition, LD spectra showed the presence of β-strand structures in the intermediate state, disclosing their orientations on the membrane surface. Furthermore, fluorescence spectroscopy showed that the formation of β-strand oligomers destabilized the membrane structure and induced the leakage of calcein molecules entrapped in the membrane. These results suggest that the formation of β-strand oligomers of M2 plays a crucial role in the disruption of the cell membrane.

## 1. Introduction

Antimicrobial resistance (AMR) is one of the most serious global threats to healthcare and agriculture. The overuse and misuse of antimicrobial agents, such as antibiotics, can lead to the emergence of antimicrobial-resistant microorganisms, resulting in prolonged hospitalizations and increased medical costs [1]. Antimicrobial peptides (AMPs) are a potential solution to this problem because they have broad-spectrum antibacterial, antifungal, and antiviral activities and it is difficult for microorganisms to acquire AMR against them [2,3]. The antimicrobial mechanism of AMP is different from that of conventional antibiotics; antibiotics inhibit the synthesis of bacterial components, whereas AMPs interact with bacterial cell membranes and directly cause damage to the membrane structure [2,4]. Extensive efforts have been devoted to use AMP as a medicine; however, the clinical application of AMP is currently limited due to its short half-life and toxic side effects [5,6,7]. Hence, further understanding of the interaction mechanism between AMP and membranes at the molecular level are necessary to gain new insights into the design strategy of effective AMP [8].

Magainin 2 (M2), a cationic and amphipathic peptide composed of 23 amino acids (GIGKFLHSAKKFGKAFVGEIMNS), is an AMP present in the immune system of African clawed frog *Xenopus laevis* [9]. The interactions between M2 and the membrane have been investigated extensively through various biophysical techniques such as fluorescence spectroscopy, circular dichroism (CD) spectroscopy, nuclear magnetic resonance (NMR) spectroscopy, and isothermal titration calorimetry (ITC) [10,11,12,13]. Regarding the molecular mechanism of the antimicrobial activity of M2, Matsuzaki et al. suggested that the M2 peptides assembled and transformed the oligomers in phosphatidylglycerol (PG) membranes, and their oligomerization induced pore formation in the membrane, leading to membrane disruption by M2 [14,15]. Schümann et al. [16] and Gregory et al. [17] found that no peptide oligomerization occurs when M2 interacts with 1-palmitoyl-2-oleoylphosphatidylglycerol phospholipids (POPG) membranes and with the mixed membranes of 1-palmitoyl-2-oleoylphosphatidylcholine (POPC) and POPG, suggesting that in a peptide state (no oligomerization), M2 works in a chaotic or stochastic manner on the pores on these membranes when obtaining the antimicrobial activity [18]. These results indicate that the mechanisms of the membrane interaction or the antimicrobial activity of M2 might strongly depend on the constituents of the lipid membrane [19]. Although these studies commonly support that M2 peptides tightly interact with negatively charged lipid membranes, such as PG, compared to neutral lipid membranes, these interactions remain to be further explored [10,14]. As for the conformation of the M2 peptide on the membrane, most reports concluded that the membrane-bound conformation of M2 was an α-helix-rich structure, but some groups, including Hirsh et al. showed that there are two populations of M2 peptides in the membranes of dipalmitoyl-phosphatidylglycerol (DPPG) and in the mixed membranes of DPPG and dipalmitoyl-phosphatidylcholine (DPPC), with one population being completely α-helical and another being completely β-strand [20,21,22], as observed in other AMPs, such as bovine lactoferricin, protegrin 1, and human β-defensin-3 [23]. These results suggest that M2 could also form both α-helix and β-strand structures on the membrane, although the contribution of these M2 conformations to the antimicrobial activity is unclear and still controversial.

In this study, we characterized the unique and complicated conformations of M2 peptides on the negatively charged DPPG membrane and their contributions to the antimicrobial activity. We measured the synchrotron-radiation circular dichroism (SRCD) and linear dichroism (LD) spectra of M2 in the DPPG liposome in the lipid-to-peptide (L/P) molar ratio from 0 to 26 and analyzed the types of secondary structures (α-helix and β-strand) of M2 and their orientation on the membrane. SRCD has been applied to characterize the secondary structures of membrane-bound proteins in various liposomes [24,25,26] and can measure the CD spectra down to the vacuum-ultraviolet region (~160 nm), whose wide range spectra could realize the component analyses of the unique and complicated conformations of M2 induced by the interaction with the DPPG membrane. LD spectroscopy can provide important information on the secondary structure orientation, disclosing the direction of the helical and strand axes of M2 against the liposome surface [24]. Furthermore, the fluorescence anisotropy of DPPG liposomes in the presence of M2 was measured to elucidate the effect of M2-membrane binding on the structure and stability of lipid membranes. The combination of SRCD, LD, and fluorescence spectroscopy could be useful for characterizing the relationships between the unique conformations and activities of M2 on the DPPG membrane.

## 2. Materials and Methods

### 2.1. Materials

The M2 peptide was synthesized in GL Biochem (Shanghai, China), GenScript Biotech Corporation (Piscataway, NJ, USA), and Bio-Synthesis (Lewisville, TX, USA). This peptide was purified by HPLC (>95%) and its molecular weight was analyzed by mass spectroscopy. 1,2-dipalmitoyl-sn-glycero-3-phospho-(1′-rac-glycerol) (sodium salt) was purchased from Avanti (Alabaster, AL, USA). 1,2-dipalmitoyl-sn-glycero-3-phosphatidylethanolamine was obtained from Cayman Chemical (Ann Arbor, MI, USA). All other chemicals were analytical grade products purchased from Sigma (St. Louis, MO, USA).

### 2.2. Liposomes and Sample Preparation

DPPG liposomes with a diameter of 100 nm were prepared by an extrusion technique, as described previously [25,27]. The DPPG lipid molecule was dissolved in 10 mM phosphate buffer (pH 7.0), and the suspension was vortexed vigorously at least above the phase-transition temperature (~41 °C). The solution was alternately placed in liquid nitrogen and heated five times (freeze-thaw cycles). The lipid suspension was then passed through a 100 nm polycarbonate membrane (Whatman, Clifton, NJ, USA) 25 times using a Mini-Extruder (Avanti). This step was performed above the phase-transition temperature. The obtained liposome vesicles were mixed with the M2 solution at a final M2 concentration of 300 μM. The concentration of M2 was determined by measuring the absorption spectra of M2 in the absence of liposomes (molar extinction coefficient at 205 nm: 94,390 M^−1^ cm^−1^). The concentration of DPPG phospholipid was changed from 0 to 8 mM (i.e., the L/P molar ratio of 0–26). The peptide-liposome mixtures were incubated overnight at 25 °C prior to the CD and LD measurements. For fluorescence anisotropy measurements, the following steps were performed before dissolving the lipids in the buffer: DPPG and DPH were dissolved in chloroform, and the stock solutions were then mixed at a DPPG/DPH molar ratio of 400/1. To remove the organic solvent, the solution was placed under a stream of nitrogen gas and then under vacuum for at least three hours [24]. The obtained DPPG film was dissolved in the buffer at final concentrations of 400 μM and 1 μM of DPPG and DPH, respectively.

### 2.3. Synchrotron-Radiation Circular Dichroism

The SRCD spectra of M2 in the presence and absence of DPPG liposomes were recorded from 260 to 178 nm using a vacuum-ultraviolet circular dichroism (VUVCD) spectrophotometer at Hiroshima Synchrotron Radiation Center (HiSOR) and an assembled optical cell with a 50 μm path length Teflon spacer. The details of the optical cell and spectrophotometer have been reported in previous studies [28,29]. The distance between the optical cell and the window of the photomultiplier tube was set to less than 10 mm to minimize the effect of light scattering from the liposome particles [24]. The temperature of the cell was controlled using a Peltier device. The actual temperature of the sample solution was obtained by a calibration curve of the actual temperature (in the optical cell) against the set temperature, which was investigated in advance. The SRCD spectra of each sample were measured four times and averaged. The SRCD spectra of the liposome solutions were also measured as a background and subtracted from the spectra of the peptide-liposome suspensions. The secondary structures of M2 in the presence and absence of DPPG liposomes were analyzed using the SELCON3 program [30,31] and the VUVCD dataset obtained by Matsuo et al. [32,33], as described previously [24,32,33].

### 2.4. Synchrotron-Radiation Linear Dichroism

The LD spectra of M2 in the presence of liposomes were measured from 300 to 185 nm using an LD spectrometer installed in the VUVCD spectrophotometer at HiSOR. The system of the LD spectrometer is described in detail elsewhere [24]. An LD flow cell with a light path length of 75 μm (Translume, MI, USA) was located within 10 mm of the photomultiplier tube. A shear flow with a flow velocity of 1.0 mL min^−1^ was applied using a dual-plunger parallel-flow pump (LC-20AD, Shimadzu, Kyoto, Japan), and the sample solution was circulated between the sample container and the LD flow cell. All LD spectra were recorded at room temperature (25 °C) with four accumulations. The scattering contribution to the LD signal was subtracted using the method described by Nordh et al. [34].

### 2.5. Fluorescence Anisotropy

The steady-state fluorescence anisotropy of DPH in DPPG liposomes was measured using a fluorescence spectrophotometer (FP-8300, Jasco, Japan) [35,36]. The excitation and emission wavelengths were 357 and 430 nm, respectively, with 5.0 nm band width. The measurements were conducted in the temperature range of 25–55 °C with 0.1 °C steps. The fluorescence anisotropy (*r*) was calculated using the following equation:(1)$$r=\frac{I_{\mathrm{vv}}-GI_{\mathrm{vh}}}{I_{\mathrm{vv}}+2GI_{\mathrm{vh}}},$$
where *I*_vv_ is the intensity of the detected light with both excitation and emission polarizers mounted vertically, *I*_vh_ is the intensity of the detected light with the excitation polarizer installed vertically and the emission polarizer mounted horizontally. The factor *G* in Equation (1) for each sample was determined by measuring the intensities of the detected light, *I*_hv_ and *I*_hh_, with the emission polarizer installed horizontally and vertically, respectively, under the excitation polarizer mounted horizontally and calculating *I*_hv_/*I*_hh_ (=*G*). The curves of the temperature dependence of anisotropy were fitted using:(2)$$r\left(T\right)=r_{l}+m_{l}\left(T-T_{l}\right)+\frac{r_{g}-r_{l}+m_{g}\left(T-T_{g}\right)-m_{l}\left(T-T_{l}\right)}{1+\exp\left(-\frac{\Delta H}{RT}\left(1-\frac{T}{T_{m}}\right)\right)},$$
where *T* is the absolute temperature, *T*_l_ and *T*_g_ are the reference temperatures for DPPG liposome in the liquid-crystalline phase (55 °C) and the gel phase (25 °C), respectively; *r*_l_ and *r*_g_ are the values of fluorescence anisotropy at 55 °C (liquid-crystalline phase) and 25 °C (gel phase), respectively; *T*_m_ is the midpoint temperature of the phase transition of the DPPG membrane; Δ*H* is the molar enthalpy change of the phase transition at *T*_m_; and *R* is the gas constant. *m*_l_ and *m*_g_ in Equation (2) are temperature dependences of the fluorescence anisotropy of lipid vesicles in the liquid-crystalline and gel phases, respectively, which are empirical parameters. The adjustable parameters in Equation (2) are Δ*H*, *T*_m_, *m*_l_, and *m*_g_.

### 2.6. Calcein Leakage Assay

M2-induced leakage of calcein entrapped in DPPG liposomes was investigated at 25 °C, as described in a previous study [10]. The fluorescence intensity of calcein was monitored at 515 nm using a fluorescence spectrometer (FP-8300, Jasco, Tokyo, Japan) with an excitation wavelength of 490 nm. The extent of calcein release was calculated according to *F* = (*I*_f_ − *I*_0_)/(*I*_max_ − *I*_0_), where *I*_0_ and *I*_f_ are the initial and final (approximately 40 min after the initial manipulation) intensities of fluorescence, respectively, and *I*_max_ is the maximal fluorescence intensity obtained by adding Triton X-100 (all the entrapped calcein was released). Calcein-loaded DPPG liposomes were prepared by the extrusion method using a solution containing 60 mM calcein and 10 mM HEPES (pH 7.0). Untrapped calcein was separated from the liposomes by gel filtration using a bio-spin chromatography column (BIO-RAD, Hercules, CA, USA) with Sephadex G-75. This step was performed above the lipid phase-transition temperature. Calcein was purchased from Tokyo Chemical Industry (Tokyo, Japan). The lipid concentration (400 μM) was determined using a phosphorus assay [37].

### 2.7. Adsorption Model Fitting Procedure

The L/P dependence of the SRCD spectra of M2 in DPPG liposomes was analyzed using an adsorption model and a global fitting algorithm. The model of adsorption for a large self-associating ligand based on scaled particle theory was used as the model for fitting in this study [38]. The model can be written as [38]:(3)$$Kc_{f}=\Phi_{1}\gamma_{1}\left(\Phi_{1},\;\Phi_{z}\right),$$
(4)$$\Phi_{z}=zK_{1z}\frac{\left(\gamma_{1}{\left(\Phi_{1},\;\Phi_{z}\right)}^{z}\right)}{\gamma_{z}\;\left(\Phi_{1},\;\Phi_{z}\right)}\;\Phi_{1}^{\;z},$$
(5)$$\ln\gamma_{1}=-\ln\left(1-\Phi_{1}-\Phi_{z}\right)+\frac{3\Phi_{1}+\left(\frac{2}{f}+\frac{1}{f^{2}}\right)\Phi_{z}}{1-\Phi_{1}-\Phi_{z}}+{\left(\frac{\Phi_{1}+\frac{1}{f}\;\Phi_{z}}{1-\Phi_{1}-\Phi_{z}\;}\right)}^{2},$$
(6)$$\ln\gamma_{z}=-\ln\left(1-\Phi_{1}-\Phi_{z}\right)+\frac{3\Phi_{z}+\left(2f+f^{2}\right)\Phi_{1}}{1-\Phi_{1}-\Phi_{z}}+{\left(\frac{\Phi_{z}+f\Phi_{1}}{1-\Phi_{1}-\Phi_{z}\;}\right)}^{2},$$
where *K* is the association constant; *c*_f_ is the concentration of the peptide free in solution (or native state); Φ_1_ (=*nc*_1_/*c*_L_) and Φ*_z_* (=*nc_z_*/*c*_L_) are the fractions of the surface area occupied by the monomer and *z*-mer, respectively; *c*_1_ and *c*_z_ are the concentrations of monomer and *z*-mer, respectively; *c*_L_ is the total lipid concentration; *n* is the number of lipid molecules covered by a single peptide; *K*_1z_ is the equilibrium constant for the formation of the *z*-mer; and *f* is the ratio of the radii of the circles representing the monomer and *z*-mer. When the area does not change due to self-association, *f* is equal to *z*^1/2^.

To fit the adsorption model to the SRCD data, the thermodynamic parameters *K*, *K*_1*z*_, *z*, and *n* were solved numerically using an iterative method. Their initial values were set, *c*_f_, *c*_1_, and *c_z_* at each L/P were calculated according to Equations (3)–(6), and then the concentration matrix *C* was output. The matrix of the pure component spectra *S* for M2 in native, monomeric, and oligomeric states was calculated using the classical least-squares approach [39]:(7)$$S={\left(C^{T}C\right)}^{-1}CD,$$
where *D* is the matrix of measured SRCD data. The 2-norm of the error *E* = |*CS* − *D*|_2_ was minimized by repeating the calculation with different initial values of *K*, *K*_1*z*_, *z*, and *n*. The obtained solutions for the parameters were then used to calculate the fractions of M2 in the three states at each L/P using the least-squares method [39]:(8)$$C=D(S^{T}{\left(SS^{T}\right)}^{-1}).$$

The populations of M2 in the three states at each L/P obtained using Equation (8) were compared with the fitting curves calculated using the optimized thermodynamic parameters. The optimized component spectra of M2 in the three states were calculated using Equation (7) with the optimized parameters.

## 3. Results

### 3.1. Synchrotron-Radiation Circular Dichroism

The SRCD spectra of M2 in the presence of DPPG liposomes (pH 7.0 and 25 °C) were measured from 260 to 178 nm in the L/P ratios from 0 to 26, as shown in Figure 1. The spectrum at L/P = 0 exhibited a negative shoulder around 220 nm, a negative peak around 200 nm, and a negative sign around 178 nm, indicating that M2 formed a random coil structure in the native state. As the L/P ratio increased, the spectral intensities of the shoulder and peak observed in the native state decreased and increased, respectively. The spectral change was completely saturated around L/P = 15, and the spectrum at L/P = 26 had two successive negative peaks around 222 and 208 nm and a positive peak around 190 nm, indicating that M2 formed a helical structure in the membrane-bound state. Furthermore, we found two iso-dichroic points around 200 and 210 nm during the spectral change, implying that M2 formed a helical conformation in the DPPG membranes through an intermediate state. M2 in the mixed liposomes with a DPPE/DPPG molar ratio of 3/1, which is believed to mimic the cell membranes of bacteria [40,41], also exhibited the same spectral change at 25 °C as those of M2 in DPPG liposomes (Appendix A), suggesting that M2 also has an intermediate state when interacting with the bacterial cell membrane.

We analyzed the spectral dataset in Figure 1 using the singular value decomposition (SVD) method. SVD, which is closely related to a principal component analysis, is a useful method for determining the dimension of a data matrix or the number of component spectra in the dataset [42,43]. As a result, the SVD analysis provided three singular values that were significantly larger than zero and estimated three component spectra that can construct the SRCD dataset within experimental error (Appendix A). This indicates that the SRCD spectra over the entire range of the L/P ratio can be explained by only three components, meaning that the spectral set in Figure 1 can be explained by the three-state model. However, the component spectra obtained from SVD might not correspond to that of M2 in each independent state (native, intermediate, and membrane-bound states) because the spectrum of M2, even in the native state, is composed of the components of secondary structures [43]. Hence, further analyses were conducted as follows:

Several physical models can explain the adsorption of peptides on membrane surfaces [38,44,45]. Among them, the adsorption model with a large self-associating ligand based on scaled particle theory successfully explained the sigmoidal shape of the adsorption isotherm of lysozyme (or ligand) on negatively charged membrane surfaces, showing that the lysozyme gathered upon the membrane association [44]. The sigmoidal shape of the adsorption isotherm is also a key behavior for interpreting the interaction between M2 and negatively charged PG because the same sigmoidal curve was observed in the tryptophan fluorescence experiments for M2 analogs and PG liposomes [14]. Hence, to characterize the intermediate state of M2, we analyzed the SRCD spectral data (Figure 1) using the adsorption model and the global fitting simulation (fitting procedures are provided in Section 2.7). This model assumes that peptides are in equilibrium between only three states: the native state in aqueous solution, the membrane-bound monomeric state, and the membrane-bound *z*-meric (oligomeric) state.

Figure 2a shows the L/P dependence of the fractional populations of M2 in the native, membrane-bound monomeric, and oligomeric states. The inset of Figure 2a shows the plots and fitting curve of CD at 193 nm against L/P. These experimental data were reproduced well from the adsorption model with a large self-associating ligand. We also found that the fitting error obtained using the adsorption model (*E* = 9.5) was much smaller than that obtained using the simplest monomodal adsorption model [44] (*E* = 16.5), which also supports the presence of the intermediate state. To understand the conformation of M2 in the intermediate state, we computed the spectra of M2 in the three states, as shown in Figure 2b. Native M2 exhibited a negative CD peak at 200 nm, which is a characteristic peak of the random coil structure [46], whereas M2 in the membrane-bound monomeric state showed two negative peaks at 208 and 222 nm, and a positive peak around 193 nm, which are characteristic of the α-helix structure [46]. Moreover, M2 in the oligomeric state exhibited a negative peak around 225 nm, and a positive peak at 200 nm, which are characteristic peaks of the β-strand structure [46], indicating that the α-helix monomers of M2 in DPPG membranes self-associate and transform to β-strand oligomers in the intermediate state.

In this model, the membrane-bound monomeric state of the peptide is modeled as a spherical form [38]. However, the helix structure of M2 might need to be treated in a cylindrical form when considering the NMR structure of M2 bound to dodecylphosphocholine micelles [47]. In fact, when Zuckermann and Heimburg [45] described the equilibrium between the peptides adsorbed on membranes and the peptide oligomers inserted into the membranes, they proposed an SPT-based model in which the membrane-bound peptide was modeled as a cylindrical form. This model might improve the fitting error in our study, but it requires the detail structural parameters of cylindrical form such as the length and the radius. Since it was difficult to obtain these structural parameters of random coil, α-helix, and β-strand structures of M2 in this study, we used the spherical form. In fact, this form reproduced well the fractional population of the three states obtained from the global fitting simulation (Figure 2).

Previous research suggested that light scattering due to the self-association of peptides and liposomes induces the distortion of CD spectra [48], and the presence of an intermediate state in the research was attributed to the effect of light scattering distortion. We measured the optical density of the M2-DPPG liposome system at 450 nm using a commercial absorption spectrophotometer and a quartz cuvette with an optical path length of 1 cm, and observed slight light scattering in the system at an L/P ratio of 3 (Appendix A). Thus, we optimized the measurement system, such as the positioning of the sample holder and detector, because this optimized system realized the measurements of undistorted CD spectra of proteins even in the presence of large unilamellar vesicles [24,49]. We observed that the raw CD spectra of M2 exhibited no baseline shift at 260 nm, which is believed to be an artifact of light scattering [48] (data not shown). These results indicate that the effect of light scattering on the SRCD spectra was negligible. Therefore, the intermediate state of M2 in DPPG liposomes does not originate from scattering artifacts.

The spectra of M2 in the three states were analyzed using the SELCON3 program to estimate the secondary structures of M2 in the native, membrane-bound monomeric, and oligomeric states. The analytical results from the CDSSTR and BeStSel programs [30,50] were almost consistent with those of SELCON3 (Appendix A). Figure 3 shows the secondary structure contents of M2 in the three states. The SELCON3 analysis revealed that native M2 comprised 7.5% α-helix, 28.4% β-strand, and 64.1% turn and unordered structures (others), showing that M2 forms a random coil in an aqueous solution. In contrast, M2 in the membrane-bound monomeric and oligomeric states included 70.2% and 1.2% α-helix, 10.9% and 46.3% β-strand, and 18.9% and 52.5% others, respectively. These data suggest that the oligomeric state of M2 interacting with DPPG membranes induced a large number of β-strand structures.

### 3.2. Synchrotron-Radiation Linear Dichroism

We then conducted LD measurements of M2 in the presence of DPPG liposomes around L/P ratios of 4 and 25, which correspond to the experimental conditions mainly occupied by the oligomeric state and the membrane-bound monomeric state, respectively, to estimate the orientation of the secondary structures (α-helices and β-strands) of M2 on the DPPG membrane surface. To obtain the LD spectra of M2, we used a liquid circulation system in which liposomes can deform from spherical to elliptical shapes in a shear flow environment. In the flow system, the average ratio of major axis/minor axis for the typical liposome, dimyristoyl phosphatidylcholine liposome, is approximately 1.7 at a flow velocity of 1.0 mL min^−1^. [24,51]. Because the long axis of the elliptical liposome was oriented along the flow direction, the average orientations of secondary structures in the liposomes showed slight biases, providing the LD signals depending on the secondary structure orientations against the membrane surface. M2 in DPPG liposomes at a flow velocity of 1.0 mL min^−1^ showed a positive peak around 195 nm with a shoulder around 205 nm at L/P = 25 and a positive peak around 200 nm and a small shoulder around 220 nm at L/P = 4, as shown in Figure 4. Furthermore, both spectra depended on the flow velocity from 0 to 1.0 mL min^−1^ (the insets of Figure 4). The observations of LD signals mean that the helical and strand structure directly interact with the DPPG liposomes. Since M2 forms helical and strand structures around L/P = 25 and 4, respectively (Figure 1) and free peptides in aqueous solution show no LD signals, these results suggest that the observed LD was mainly affected by the oriented helical structure of M2 for L/P = 25 and the oriented strand structure for L/P = 4 on the elliptical liposomes.

The electric or magnetic dipole moments of helical structures are approximately vertical to the helix axis at 190 nm, parallel at 208 nm, and vertical at 222 nm [51]. Furthermore, the absorption spectrum of M2 at an L/P ratio of 25 was successfully reproduced by the three component spectra of Gaussian functions, which have peak positions around 222, 208, and 190 nm, respectively, as shown in Appendix A. These results allowed us to analyze the LD spectra under two different assumptions: first, all M2 peptides form the same helical orientation with a single helical angle against the membrane surface; second, M2 peptides form perpendicular (transmembrane) and parallel helical structures against the membrane surface. The results under the first assumption showed that the LD spectrum could be interpreted as the angle between the M2 helix axis and the membrane normal (48°) (Appendix A) while those under the second assumption showed that the ratio of M2 helix axes perpendicular and parallel to the membrane surface was 1:1.2 (Appendix A). Both fitting analyses reproduced the experimental LD well, as shown in Appendix A. As for the LD spectrum at an L/P of 4, the electric or magnetic dipole moments of the β-strand structure were approximately vertical to the strand axis at 195 nm, parallel at 219 nm, and vertical at 221 nm [51,52]; however, the fitting analysis, as described in Appendix A, was difficult because the two dipole moments (219 and 221 nm) were very close and had inverse directions, which induced cancelation. However, a positive peak around 200 nm was clearly detected in the LD spectrum and showed that the axis of the β-strands of M2 was perpendicular to the membrane surface on average.

### 3.3. Fluorescence Anisotropy

To investigate the contribution of the α-helix formation in the monomeric state and the β-strand formation in the oligomeric state to the stability of lipid membranes, we measured the fluorescence anisotropy of DPPG liposomes in the presence of M2 at L/P = 25 and 3. Fluorescence anisotropy can monitor the degree of dynamics or rotational mobility of DPH in liposomes, and the parameters are then used to investigate the lipid packing (ordering) or the membrane fluidity of lipid bilayers [35,53]. The DPH probe is present in the hydrophobic core of liposomes and the motion of the DPH molecules is normally restricted to the gel-phase liposomes, resulting in high fluorescence anisotropy values. However, when the temperature is increased or the phase of the lipid bilayer shifts to the liquid-crystalline phase, the degree of rotational mobility of the DPH molecules increased, leading to a decrease in the anisotropy values. Figure 5 shows the temperature dependence of the fluorescence anisotropy values *r* of DPPG liposomes in the absence of M2 and at L/P = 25 and 3.

As shown in the inset of Figure 5, the *r* at 25 °C linearly decreased as L/P decreased, up to 10, whereas *r* did not change below a L/P of 10, implying that the effect of M2 binding on the DPPG membrane changes around L/P = 10. We analyzed the midpoint temperature *T*_m_ and the molar enthalpy change Δ*H* of the phase transition of DPPG liposomes in the presence of M2 using a sigmoid function (Equation (2)). The fitting curves were well fitted to the experimental data (Figure 5). The *T*_m_ and Δ*H* values are listed in Table 1. From this table, pure DPPG liposomes showed a phase-transition temperature (*T*_m_) of approximately 40.1 °C, which is consistent with the main phase-transition temperature of the DPPG membrane (*T*_m_ = 41 °C), showing that the DPH fluorescence anisotropy measurement can successfully monitor the change in membrane fluidity of DPPG liposomes. The *T*_m_ of DPPG liposomes at a L/P = 25 was 41.7 °C, which was higher than that of pure DPPG liposomes. In contrast, the *T*_m_ of DPPG liposomes at L/P = 3 was 31.3 °C, showing a significant decrease. This indicates that M2 in the DPPG membranes at L/P = 3 destabilizes the bilayer structure of the membranes. Since M2 can form a monomeric helical structure and oligomeric strand structure at L/P = 25 and 3, respectively, these findings suggest that the adsorption of monomeric and oligomeric M2 on the DPPG membranes largely contributes to the stabilization and destabilization of the membrane, respectively.

## 4. Discussion

In this study, we used SRCD spectroscopy to reveal that the conformation of M2 changed from random coil to α-helix structures via an intermediate state as the L/P ratio increased. The global fitting analysis of the SRCD spectra based on the adsorption model indicated that α-helical M2 monomers assembled and transformed into β-strand M2 oligomers in the intermediate state. LD data also showed that the β-strand M2 oligomers bind to the membrane as the strand axis is perpendicular to the membrane surface. To elucidate the effect of oligomerization on the structure and stability of lipid membranes, further we conducted fluorescence anisotropy measurements of DPPG liposomes in the presence of M2 and revealed that the formation of an oligomeric β-strand structure contributed to the destabilization of the membrane structure. Therefore, our findings demonstrate that the oligomeric β-strand structure of M2 in membranes plays a crucial role in the disruption of the cell membrane.

Our SRCD data showed that M2 formed a helical conformation on the membrane at L/P = 25 and in the monomeric state. The LD result at L/P = 25 could be interpreted as the two possible orientations of M2 with respect to the membrane surface: the helix axis of M2 for the membrane normal was uniformly distributed at an angle of 48° or the helix axes of M2 perpendicular and parallel to the membrane surface were mixed at a ratio of 1:1.2. In addition, our fluorescence anisotropy data showed an increase in the membrane stability at L/P = 25. According to previous research [40], pure electrostatic interactions between negatively charged DPPG headgroups and positively charged residues of peptides contributes to the increase in membrane stability, whereas the hydrophobic interaction of peptides with membrane core regions induces perturbation of lipid chain packing, leading to the destabilization of the membrane structure. Furthermore, Fourier transform infrared (FTIR) and solid-state NMR spectroscopy have indicated that α-helical M2 analogs directly bind to the headgroups of DPPG phospholipids [21]. If the helix axis of M2 was uniformly distributed at an angle of 48° against the membrane normal, M2 would be partially inserted into the membrane, resulting in hydrophobic interactions, destabilizing the membrane structure. Wimley [19] suggested four types of models for the molecular mechanism of the antimicrobial activity of M2, one of which is that M2 peptides can form a transmembrane pore. In this case, the polar faces of peptides whose helical orientation is perpendicular to the membrane surface can interact with the polar headgroups of phospholipids [7], stabilizing the membrane structure. Hence, it is possible that helical M2 interacts with DPPG headgroups and exists in a mixture of perpendicular and parallel helices. Furthermore, because the toroidal pore formation and other models, such as carpet and detergent models do not require specific peptide-peptide interactions [19], it is also possible that M2 peptides in the pore formation could be treated as a monomeric state.

To confirm the presence of pore formation at L/P = 25, we tested M2-induced leakage of calcein molecules entrapped in DPPG liposomes and found that approximately 30% of the dye molecules were released from the liposomes (Appendix A). This result indicates that the pores formed in liposomes at L/P = 25, but the M2-induced disruption of DPPG liposomes would not be completed under this condition. In contrast, the extent of liposome-entrapped calcein leakage at L/P = 4 was 2-fold greater (~60%) than that at L/P = 25. Hence, elucidating what occurred around L/P = 4 might be important for understanding the overall molecular mechanism of DPPG liposome disruption by M2.

We found that the α-helix monomers of M2 assembled and transformed into β-strand oligomers as the L/P ratio decreased from 25 to 4. Although our fitting analysis of the SRCD data did not yield the unique thermodynamic parameters of the M2 adsorption to DPPG membranes due to strong cross-correlation between each parameter and the limited number of data points (Appendix A), the fitting analysis suggested that M2 in the DPPG membranes formed the β-strand *z*-mer (*z* > 2). We also estimated that the average axis of β-strands was perpendicular to the membrane surface and confirmed that the formation of β-strand oligomers contributed to the destabilization of the membrane structure. Since the hydrophobic interaction of peptides with membrane core regions induces destabilization of the membrane structure [40], the β-strand oligomers of M2 may be buried in the DPPG membrane core or the disruption of the chain packing of DPPG membranes. These results suggest that the oligomeric β-strand M2 has unique conformations, such as the toxic β-sheet channels of protegrin-1 [54] and transmembrane amyloid β (Aβ) oligomers [55]. Ciudad et al. suggested that the β-strand-rich Aβ42 tetramer formed lipid-stabilized pores in a membrane-mimicking environment, in which water permeation occurred [55]. The discussion requires further analyses such as FTIR [21], vibrational sum frequency generation [56], and Raman spectroscopy [22]. However, the β-strand oligomers of M2 may contribute to the formation of stable pores in DPPG membranes, inducing the complete disruption of the membrane structure.

Previous reports [57] suggested that Δ*H* and *T*_m_ of membrane phase transition were basically correlated with each other for several membrane-bound proteins and peptides, meaning that the destabilization of membrane (decrement in *T*_m_) would relate to the decrement in Δ*H*. In this study, we observed that both Δ*H* and *T*_m_ of the phase transition of DPPG liposome decreased at L/P = 3, compared to those of pure DPPG liposome (Table 1), which identifies with the previous results and suggests that the penetration of oligomeric M2 peptides would contribute the destabilization membrane and the decrement in Δ*H*. However, the Δ*H* and *T*_m_ at L/P = 25 decreased and increased, respectively. This conflict might be induced by the unique conformations of M2 on the membrane because M2 would form two types of orientations on the membrane (the helix axes of M2 perpendicular and parallel to the membrane surface were mixed at a ratio of 1:1.2). The Δ*H* and *T*_m_ at L/P = 25 might be also interpreted from a balance between the stabilization by the electrostatic interaction and the destabilization by the pore formation.

We investigated the conformation of M2 in dimyristoyl-phosphatidylglycerol liposomes and observed that M2 in the liposomes did not show the presence of an oligomeric state with a β-strand structure (data not shown). It is likely that the formation of oligomeric β-strand structures largely depends on the inherent characteristics of lipid constituents, such as the phase-transition temperature and tail region length. Given that bacteria change the lipid composition, including the acyl chain, of their cell membranes in response to their environment [58,59,60], to understand the antimicrobial mechanism of M2, it is important to characterize the detailed dependence of lipid constituents. In addition, it is important to investigate the interactions between M2 and bacterial cells [61] to clarify whether M2 can form β-strands when interacting with bacterial cells under physiological conditions.

## 5. Conclusions

In this study, SRCD, LD, and fluorescence spectroscopy were used to characterize the relationships between the conformations and activities of M2 on the DPPG membrane. We found that M2 formed β-strand oligomers in DPPG liposomes, which induced the destabilization of the membrane structure and the leakage of calcein molecules entrapped in the membrane. It has been commonly recognized that the formation of the helical structure of M2 on the membrane could be the driving force of the destabilization of the membrane or the antimicrobial activity; however, our findings demonstrate that the oligomeric β-strand structure of M2 in membranes also plays a crucial role in the disruption of the cell membrane, and future studies will be necessary to characterize the molecular mechanisms underlying the antimicrobial action of M2, which would be helpful for gaining new insights into the antimicrobial mechanism of AMPs.

## Figures and Tables

**Figure 1 membranes-12-00131-f001:**
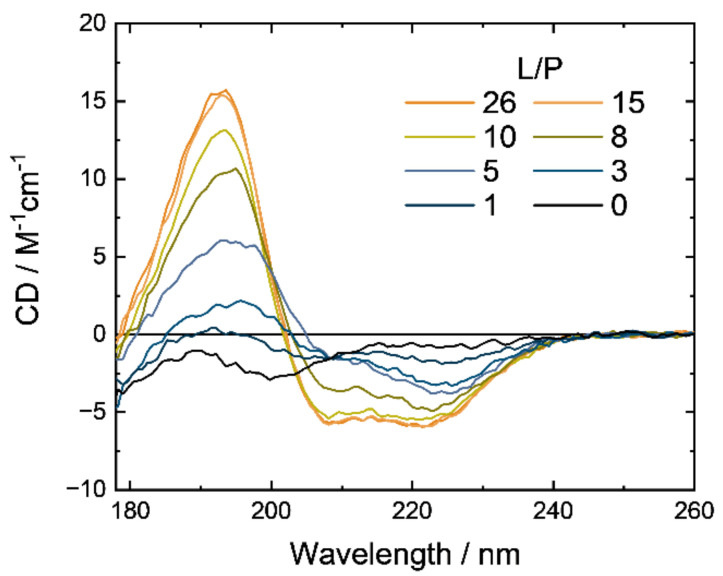
Synchrotron-radiation circular dichroism (SRCD) spectra of M2 in the presence of DPPG liposome in the L/P ratio from 0 to 26. All spectra were recorded at 25 °C, pH 7.0, and the M2 concentration of 300 μM. Experimental conditions are described in detail in Section 2.3.

**Figure 2 membranes-12-00131-f002:**
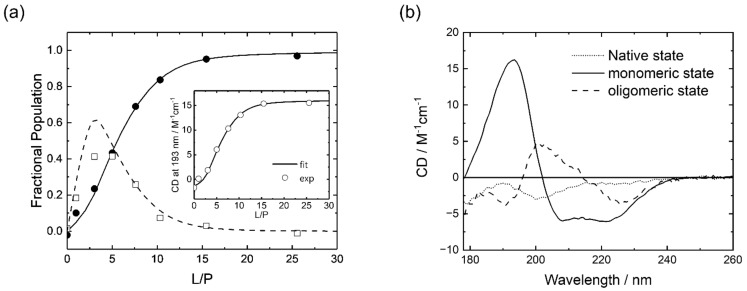
(**a**) Lipid-to-peptide (L/P) dependence of fractional populations of M2 in membrane-bound monomeric (closed circle: plots; solid line: fitting curve) and oligomeric states (open square: plots; dashed line: fitting curve) (see Section 2.7 for the details of calculations). The fitting curves in (**a**) were calculated using the following optimized parameters: *K* = 17500, *K*_1*z*_ = 3, *z* = 5, and *n* = 2.2. The inset of (**a**) shows the L/P dependence of the CD at 193 nm; (**b**) spectra of M2 at native, membrane-bound monomeric, and oligomeric states.

**Figure 3 membranes-12-00131-f003:**
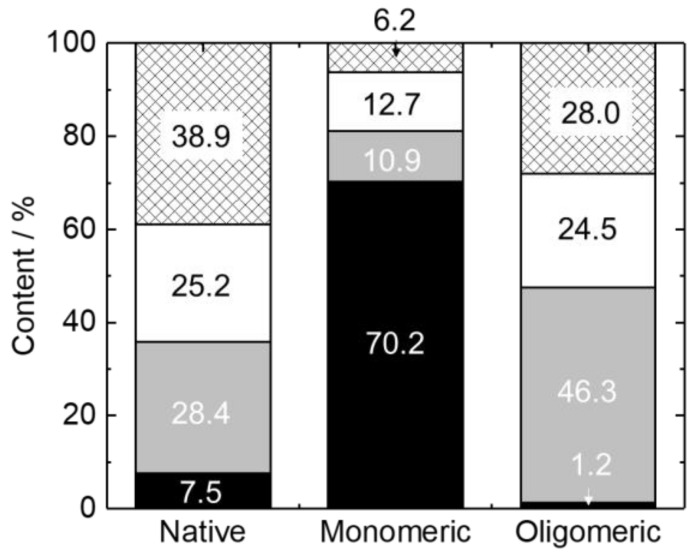
Bar graphs of secondary structures of M2 in native, membrane-bound monomeric, and oligomeric states analyzed using the SELCON3 program (black: α-helix; gray: β-strand; white: turn; meshed: unordered structure).

**Figure 4 membranes-12-00131-f004:**
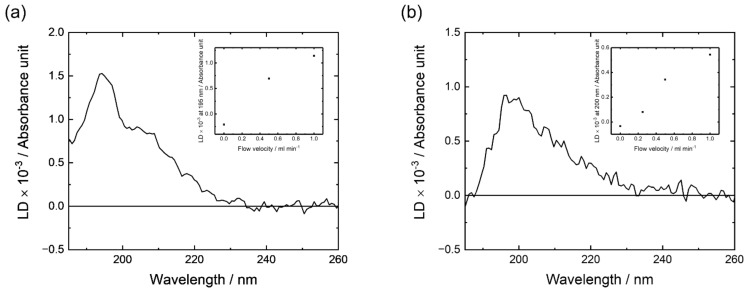
Linear dichroism (LD) spectra of M2 in DPPG liposomes at the lipid-to-peptide ratio (L/P) of (**a**) 25 and (**b**) 4. The insets of (**a**,**b**) show the dependence of flow velocity on the LD at 195 and 200 nm, respectively. All spectra were recorded at 25 °C, pH 7.0, and M2 concentration of 200 μM. Experimental conditions are described in detail in Section 2.4.

**Figure 5 membranes-12-00131-f005:**
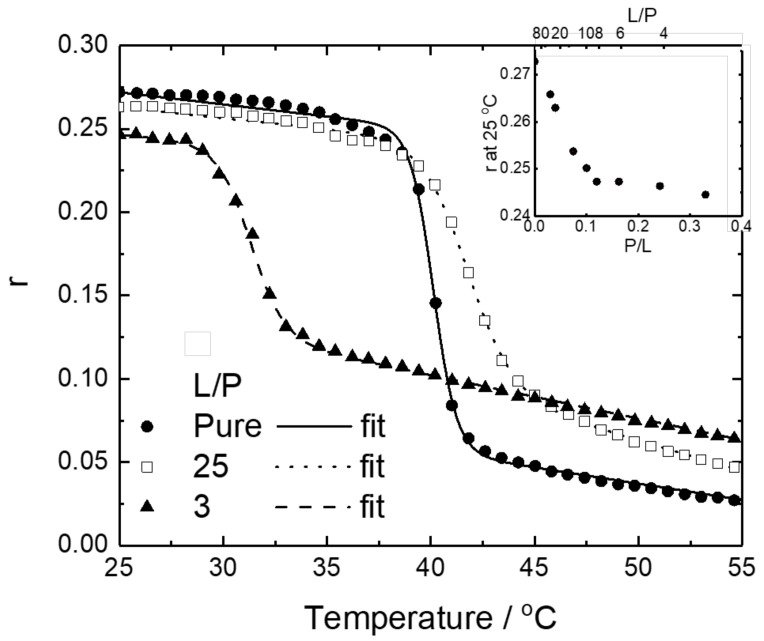
Fluorescence anisotropy values *r* and fitting curves of DPPG liposomes in the absence of M2 and in the presence of M2 at L/P = 25, and 3. The inset shows P/L (L/P) dependence of *r* at 25 °C. All data were recorded at pH 7.0 and DPPG concentration of 400 μM. Experimental conditions are described in detail in Section 2.5.

**Table 1 membranes-12-00131-t001:** Midpoint temperature *T*_m_ and molar enthalpy change Δ*H* obtained by fitting the temperature dependence of the fluorescence anisotropy in the absence of M2 and in the presence of M2 at L/P of 25 and 3 in Figure 5.

L/P	*T*_m_ [°C]	Δ*H* [kcal/mol]
Only liposome	40.12 ± 0.02	351 ± 7
25	41.71 ± 0.02	180 ± 4
3	31.31 ± 0.03	231 ± 7

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
