# Peer review of "Formation of β-Strand Oligomers of Antimicrobial Peptide Magainin 2 Contributes to Disruption of Phospholipid Membrane"

_membranes, 2022, doi:10.3390/membranes12020131_

Round 1

Reviewer 1 Report

Please find an attached file of comments

Author Response

Answer to Reviewer 1

Thank you very much for the valuable comments and for giving us the opportunity to revise our manuscript. All revisions were marked up using the “Track Changes” function in the manuscript so that you can find the modified words and sentences easily. Below are point-by-point answers to the questions and comments. We hope that these revisions are satisfactory and the revised version will be acceptable for publication in Membranes.

Reviewer 1

1: Line 17~18 (Abstract)

Thank you very much for considering the expression of this phrase. We carefully checked your comments and confirmed that your understandings and explains are correct. We think, however, the expression of “with increasing L/P ratio” is still correct because when we use the phrase of “with increasing P/L ratio”, abstract mentions that M2 forms the random coil structure in the low P/L ratio (condition with much amount of lipid molecules and low amount of M2) but the experiments showed the helical-rich structure, and also abstract mentions that M2 forms the helical structure in the high P/L ratio (condition with low amount of lipid molecules and high amount of M2) but the experiments showed the strand structure (as an intermediate state) or the random coil structure without lipid molecules. Hence, we would like to keep this phrase.

2: Line 346~349

We agree with your points because there are no experimental data of L/P=25 and 4 in Figure 2(a). We used the both ratios because the analytical results of SRCD spectra (Figure 2(a)) showed that the optimal L/P conditions for LD should be L/P = 3~4 for β-strand-rich condition and L/P = 25 for α-helix-rich condition (Lines 333~337 in the revised version). To describe more accurately, we would like to modify “at” to “around” in this sentence (Line 350 in the revised version).

3: We agree with your points and have revised the manuscript as you suggested (line 382, 383, 392~395, 403, 405, 406, 408, inset of Figure 5 and Table 1 in the revised version).

4: Table 1

Thank you very much for important question. Previous reports [57] suggested that ΔH and Tm of membrane phase transition were basically correlated with each other for several membrane-bound proteins and peptides, meaning that the destabilization of membrane (decrement in Tm) would relate to the decrement in ΔH. In this study, we observed that the both ΔH and Tm of the phase transition of DPPG liposome decreased at L/P = 3, compared to those of pure DPPG liposome (Table 1), which identifies with the previous results and suggests that the penetration of oligomeric M2 peptides would contribute the destabilization membrane and the decrement in ΔH. However, the ΔH and Tm at L/P = 25 decreased and increased, respectively. This conflict might be induced by the unique conformations of M2 on the membrane because M2 would form two types of orientations on the membrane (the helix axes of M2 perpendicular and parallel to the membrane surface were mixed at a ratio of 1:1.2). The ΔH and Tm at L/P = 25 might be also interpreted from a balance between the stabilization by the electrostatic interaction and the destabilization by the pore formation. These sentences were added in the manuscript (Lines 486~498 in the revised version).

5: Calcein leakage experiments

We apologize for no data of Calcein leakage. The results of the experiments were added in the supporting information as Figure S7.

6: Lines 316~317

Thank you for giving us adequate sentence. We modified this sentence as you suggested.

7: LD experiments

We added the sentences related to the shape of ellipse including the ratio of minor axis/major axis (Lines 339~341 in the revised version). It would be possible that the membrane-bound oligomers leave from membrane surface under the high flow velocity but it is strongly expected that these free oligomers have random orientations in solution, inducing no LD signal. In this study, we observed LD signal which means that the helical and strand structure directly interact with the DPPG liposomes and they are oriented corresponding to the degrees of liposome distortion due to the flow velocity. The related sentences were added in the manuscript (Lines 358~351 in the revised version).

Reviewer 2 Report

In the present study, Munehiro Kumashiro et al measured the SRCDand linear LD spectra of M2 in dipalmitoyl‐phosphatidylglycerol lipid membranes to characterize the conformation and orientation of M2 on the membrane to further interpret the membrane active action mechanism of M2. As we know, AMPs was believed to be a potential candidate of antimicrobial agents to defend bacteria, especially MDR bacteria. It is generally accepted that AMPs exert antimicrobial activity by destroying the integrity of cell membrane, so they were not affected by the traditional drug resistant mechanism. However, the peptide-membrane interaction mechanism is still unclear. So, new methods or instruments were still needed to investigate the peptide-membrane interaction mechanism from different perspective. The present study may provide a strategy for further explaining M2-membrane interaction and the design and the investigation of new antimicrobial peptides. This paper was well organized and written. I would like to recommend that this paper is accepted in the present form.

Author Response

Answer to Reviewer 2

Thank you very much for your comments and for encouraging us for the further investigation of the AMP-membrane interaction mechanism by using new techniques such as SRCD-based methods. As you mentioned, we also expect that our study would provide new insights for explaining M2-membrane interaction and for design of new antimicrobial peptides.